# A partially self-regenerating synthetic cell

Barbora Lavickova [1], Nadanai Laohakunakorn[2] & Sebastian J. Maerkl [1✉]

Self-regeneration is a fundamental function of all living systems. Here we demonstrate partial molecular self-regeneration in a synthetic cell. By implementing a minimal transcription-translation system within microfluidic reactors, the system is able to regenerate essential protein components from DNA templates and sustain synthesis activity for over a day. By quantitating genotype-phenotype relationships combined with computational modeling we find that minimizing resource competition and optimizing resource allocation are both critically important for achieving robust system function. With this understanding, we achieve simultaneous regeneration of multiple proteins by determining the required DNA ratios necessary for sustained self-regeneration. This work introduces a conceptual and experimental framework for the development of a self-replicating synthetic cell.

[1] Institute of Bioengineering, School of Engineering, École Polytechnique Fédérale de Lausanne, Lausanne, Switzerland. [2] Institute of Quantitative Biology, Biochemistry, and Biotechnology, School of Biological Sciences, University of Edinburgh, Edinburgh, United Kingdom. ✉email: sebastian.maerkl@epfl.ch

Bottom-up construction of a self-replicating synthetic cell that exhibits all the hallmarks of a natural living system is an outstanding challenge in synthetic biology[1–3]. Although this goal is ambitious, progress is rapidly accelerating, and key structures and functions required for constructing a synthetic cell, including compartmentalization[4–6], mobility and shape[7–9], metabolism[10,11], communication[12,13], and DNA replication[14–16], have recently been demonstrated, suggesting that integration of these subsystems into a functional synthetic cell may be an attainable goal.

A biochemical system able to fully self-regenerate or self-replicate, is a crucial requirement for construction of a synthetic cell. A self-replicating artificial system has been first proposed by von Neumann in the 1940s[17]. Von Neumann developed the concept of a universal constructor, which is an abstract machine capable of self-replication using a set of instructions, external building blocks, and energy. So far, universal constructors have only been implemented in silico in the form of cellular automata[18]. Similar concepts have been explored experimentally with auto-catalytic chemical systems[19] and self-replicating ribozymes[20]. A self-replicating biochemical system is strictly analogous to the universal constructor in that it would be capable of self-replication using instructions encoded in DNA while being supplied with building blocks and energy (Fig. 1a). A physical implementation of a universal constructor could therefore be theoretically achieved by a minimal recombinant transcription–translation system capable of regenerating all of its components including proteins, ribosomes, tRNAs, and DNA[21]. DNA replication has recently been demonstrated in vitro[14–16] and progress is being made in reconstituting ribosomes[22–24] and tRNAs[25]. Here, we demonstrate the principal steps towards constructing a universal biochemical constructor by creating a system capable of sustained self-regeneration of proteins essential for transcription and translation.

Development of a transcription–translation system capable of self-regeneration faces several challenges. First, synthesis capacity of the system in terms of its protein synthesis rate must be sufficient to regenerate the necessary components. This problem is exacerbated by the fact that protein synthesis capacity drastically decreases in a non-optimal system[26–28]. Second, the components being regenerated must be functionally synthesized which may require chaperones, and modifying enzymes. And third, the reaction must take place in an environment that allows continuous and sustained regeneration.

The PURE (protein synthesis using recombinant elements) system[29] is a viable starting point for achieving self-regeneration because of its minimal nature as well as its defined and adjustable composition[30]. Batch expression experiments combined with polyacrylamide gel electrophoresis (PAGE) and mass-spectrometric (MS) analysis indicated that the PURE system should be able to synthesize ~70% of all *Escherichia coli* proteins[31]. Moreover, it was recently shown that co-expression of multiple PURE components in a single batch reaction yielded the required concentrations for self-replication[15]. However, these experiments didn't determine whether proteins were functionally synthesized, which varies largely for proteins expressed in the PURE system[32,33]. Other studies showed that the 30S ribosomal subunit[23,24], and 19 of 20 aaRSs, can be functionally synthesized in the PURE system[34]. All of those experiments were performed in batch or continuous-exchange formats and self-regeneration of any component has yet to be demonstrated.

Here, we employ continuous transcription–translation reactions operating inside microfluidic reactors[35] to demonstrate self-regeneration of essential protein components. Our approach using the PURE system, microfluidic chemostats, and monitoring fluorescent protein production, allows activity and performance of self-regeneration to be assessed in real-time. We implemented a kick-start method to boot-up regeneration of essential PURE proteins from DNA templates. We demonstrate the concept and feasibility of this approach by regenerating different aminoacyl-tRNA synthetases (aaRSs). We also regenerate T7 RNA polymerase (RNAP) and map system optimality by varying T7 RNAP DNA concentration and are able to explain the observed genotype–phenotype relationships with a biophysical resource limitation model. We go on to show that several proteins can be regenerated simultaneously by regenerating up to seven aaRSs. This proof-of-principle work takes steps towards constructing a self-replicating transcription–translation system and provides a viable approach for developing and optimizing other critical subsystems including DNA replication, ribosome synthesis, and tRNA synthesis, with the goal of achieving a self-replicating biochemical constructor in the near term and ultimately a viable synthetic cell.

## Results

**Experimental design**. To maintain continuous cell-free reactions, we improved a microfluidic chemostat previously used for implementing and forward engineering genetic networks in vitro[35,36]. The device consists of eight independent, 15 nL reactors, with fluidically hard-coded dilution fractions defined by reactor geometry, as opposed to the original device, which used peristaltic pumps for metering (Fig. 1b, Supplementary Fig. 1, Supplementary Movie 1)[37]. During experiments 20% of the reactor volume was replaced every 15 min with a ratio of 2:2:1 for energy, protein/ribosome, and DNA solution, respectively, resulting in an effective dilution time of ~47 min (Fig. 1a, Supplementary Tables 1, 2). Another key improvement was the supply of multiple solutions without the need for cooling. This was achieved by storing the energy and protein components separately, which when stored pre-mixed and without cooling resulted in non-productive resource consumption[38]. Second, reaction temperature was set to 34 °C, which decreased PURE degradation with only a minor decrease in protein synthesis rate (Supplementary Fig. 2). At last, as the redox reagent used in the PURE system is known to degrade rapidly, we eliminated 1,4-dithiothreitol in the energy solution and instead added tris(2-carboxyethyl)phosphine (TCEP) to the energy and protein solutions. To allow PURE system modification and omission of protein components we produced our own PURE system based on the original formulation[26,39]. For each protein regenerated, we produced a ΔPURE system lacking that particular protein or proteins. This allowed us to validate that the omitted protein is essential for system function. We furthermore adjusted PURE protein composition by reducing the concentration of several aaRSs (Supplementary Tables 3, 4).

In all experiments we expressed a fluorescent protein (eGFP) as an indicator of functional self-regeneration and to provide a quantitative readout of protein synthesis capacity. We developed a "kick-start" method to enable the system to self-regenerate proteins from DNA templates (Fig. 1c). The experimental design involves three distinct phases: kick-start, self-regeneration, and washout. The kick-start phase is required to allow a productive switch from a complete to a ΔPURE system to occur. The self-regeneration phase tests whether the system functionally regenerated the omitted protein component or components, and the washout phase serves as a control to prove that the omitted component or components were indeed essential for system function. In the kick-start phase, which lasts for the first 4 h, linear DNA templates coding for eGFP and the protein to be regenerated are added to a complete PURE system. This leads to the expression of eGFP and the protein to be regenerated. In the

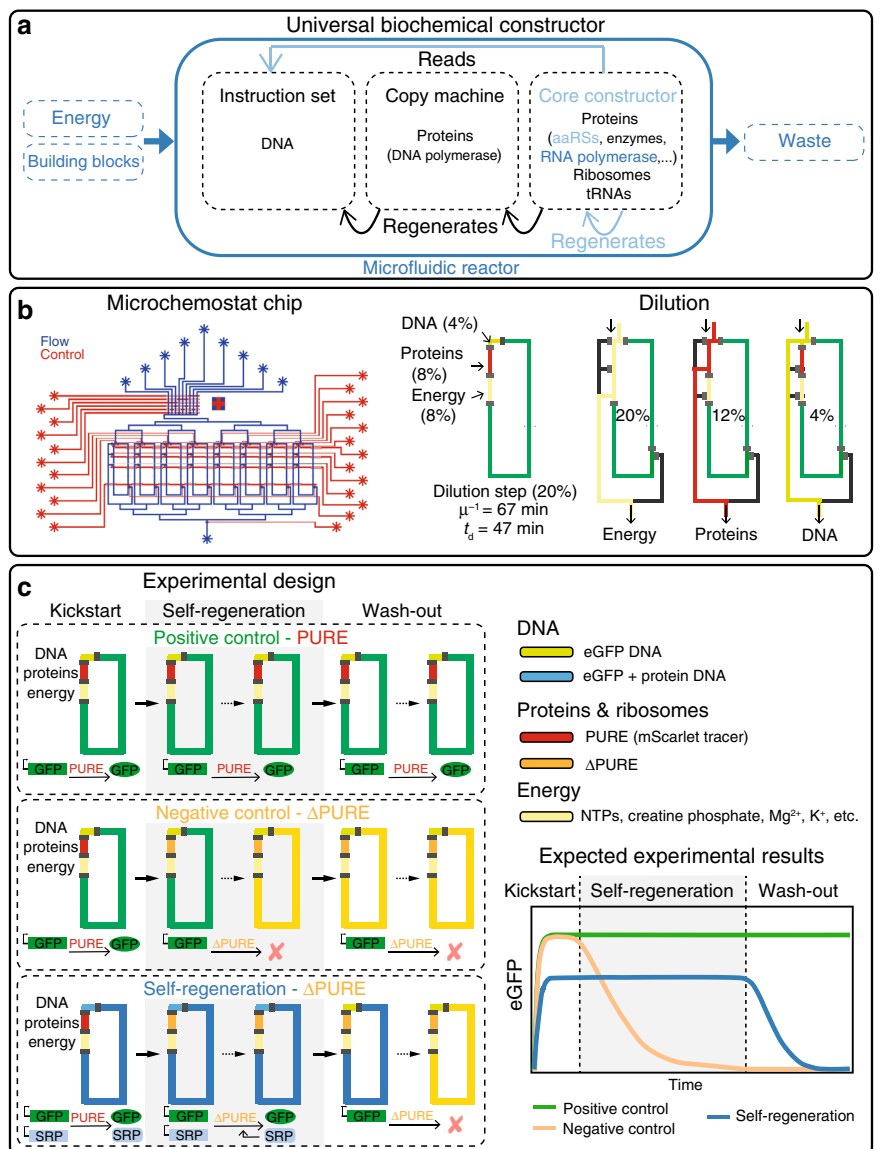

**Fig. 1 Experimental design for a self-regenerating synthetic cell. a** Diagram of the universal biochemical constructor concept. Systems, components, and functions colored in blue and light-blue were fully or partially implemented in this work, respectively. **b** Design schematic of the microfluidic device with eight individual chemostat reactors. Flow layer is shown in blue and control layer in red. Design and functional details are provided in Supplementary Fig. 1. A schematic representation of one dilution cycle where 20% reaction volume is replaced every 15 min. Dilution rate $\mu = -\ln(C_t/C_0) \cdot t^{-1}$, residence time $\mu^{-1}$ and dilution time $t_d = \ln(2) \, \mu^{-1}$. One dilution cycle consists of three steps: energy solution (yellow) is loaded via the 20% segment, protein and ribosome solution (red) is flushed through the 12% segment, and DNA solution (green) through the 4% segment, resulting in the desired composition of 8%, 8%, and 4%, respectively. **c** Experimental design, including the three experiment phases: kick-start, self-regeneration, and washout. Solutions are loaded in the rings at different time points: energy solution (yellow), full PURE (red), ΔPURE (orange), eGFP DNA (green), and eGFP + protein DNA (blue). A schematic showing the expected results for the different experimental phases indicating early cessation of synthesis activity for the negative control (yellow), continuous synthesis activity in the positive control (green), and continuous synthesis for self-regeneration (blue) during the self-regeneration phase followed by cessation of synthesis activity during the washout phase.

self-regeneration phase, the full PURE is gradually replaced with a ΔPURE solution lacking the particular protein that is to be regenerated. Thus at steady state, the system will remain functional only through self-regeneration of the omitted protein. Finally, in the washout phase, DNA encoding the protein being regenerated is no longer added to the system leading to dilution of the protein being regenerated. Once a critical concentration for the regenerated protein is reached overall protein synthesis falls and ultimately ceases.

We implemented two additional control reactions in most experiments. Positive controls use full PURE and express only eGFP during all three phases and serve as a validation of steady-state chemostat function and a reference point for maximal protein synthesis capacity of an unloaded and optimal PURE reaction. Negative controls switch between complete and ΔPURE, but do not contain DNA template for the omitted protein component. This confirms that without self-regeneration, protein synthesis activity is indeed rapidly lost. We spiked the full PURE protein fraction with an mScarlet tracer to confirm that all fluid exchanges take place and the device functioned correctly.

**Aminoacyl-tRNA synthetase regeneration**. As a proof-of-concept and validation of the experimental design, we tested

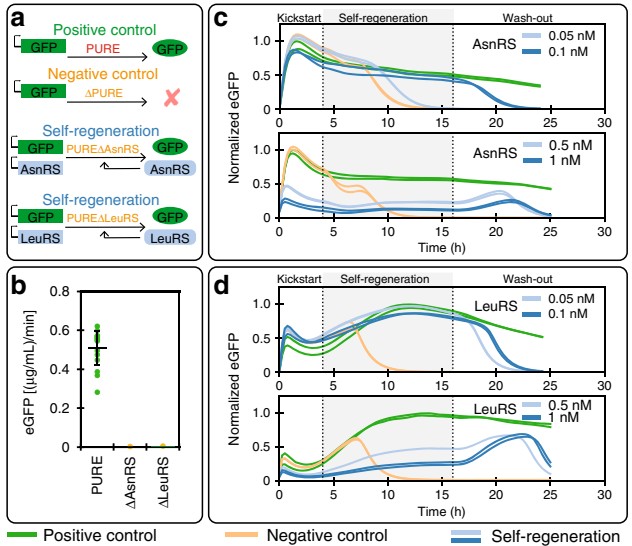

**Fig. 2 Aminoacyl-tRNA synthetase regeneration. a** Overview of the different aaRSs regeneration experiments. **b** eGFP batch synthesis rates for the full PURE system and AsnRS or LeuRS ΔPURE systems (mean ± s.d. for PURE system (green, $n = 16$ technical replicates), and mean for ΔPURE systems (yellow, $n = 2$ technical replicates)). Self-regeneration experiments for **c** AsnRS and **d** LeuRS at different DNA concentrations (positive control: green, negative control: yellow, self-regeneration: blue). Results for all DNA concentrations tested and corresponding mScarlet traces can be found in Supplementary Fig. 6. The level of eGFP is normalized to the maximum level attained in the positive control. The composition of PURE systems used are given in Supplementary Table 3, 2 nM of eGFP template was used for all experiments, aaRS DNA template concentrations are indicated. Source data are available in the Source Data file.

regeneration of two aaRSs: Asparaginyl-tRNA synthetase (AsnRS) and Leucyl-tRNA synthetase (LeuRS) (Fig. 2a). We first carried out batch experiments to ascertain synthesis of the synthetases in our PURE system (Supplementary Fig. 3). We also validated that both synthetases are essential by omitting them individually from a PURE reaction (Fig. 2b). When we used the original PURE system's aaRS concentrations, decreases in protein synthesis activity were observed only after extended washout periods because the critical aaRS concentrations were reached only after numerous dilution cycles. We therefore reduced the concentrations of the aaRSs being regenerated so that fast activity declines during washout occurred, while preserving high-protein synthesis rates (Supplementary Fig. 4, Table 3).

We achieved successful self-regeneration for both AsnRS and LeuRS and complete loss of protein synthesis activity during washout (Fig. 2c, d). We tested four DNA concentrations for each aaRSs. AsnRS and LeuRS regeneration at DNA concentrations of 0.1 nM and 0.05 nM, respectively, resulted in high system activity comparable to the positive control throughout the self-regeneration phase. If an insufficient DNA template concentration of 0.05 nM was provided for AsnRS, a decrease in eGFP fluorescence was observed identical to the negative control but with a slight delay. A twofold difference in DNA template concentration thus resulted in either optimal self-regeneration or complete system failure. For LeuRS a similar twofold change was less consequential with either concentration resulting in self-regeneration, but with slightly lower expression obtained for the higher concentration of 0.1 nM. Higher DNA concentrations resulted in robust but markedly lower system activity for both

aaRSs. These studies showed that our experimental design enables self-regeneration and that self-regeneration can be achieved with two different aaRSs.

DNA input concentration is critically important for system function. When higher than optimal DNA concentrations were used, we observed successful and robust self-regeneration, as indicated by the maintenance of synthesis activity above negative control levels, but considerably lower eGFP expression levels as compared with the positive control. Because no negative effects were observed in batch reactions for high aaRS protein concentrations in the PURE system (Supplementary Fig. 4)[40], we attribute this effect to a resource competition or loading effect between the protein being regenerated and eGFP[41]. The onset of this loading effect can be estimated by measuring the DNA concentration for which system output saturates, which is ~1 nM for the PURE system (Supplementary Fig. 5). eGFP DNA template is present at a concentration of 2 nM in all experiments and is thus fully loading the system. Any additional DNA added to the system will thus give rise to resource competition effects.

A simple resource competition model gives rise to a couple of specific predictions. First, the level of eGFP synthesized during self-regeneration should never rise above the positive control, assuming that the concentration of the self-regenerated protein is at an optimal level in the positive control. This is because synthesis of an additional protein leads to resource competition and lower eGFP levels. Low concentrations of aaRS DNA has a minimal loading effect since the ratio of aaRS to eGFP DNA is small. As the concentration of aaRS DNA is increased the loading effect becomes stronger, leading to a noticeable decrease in eGFP levels. The second prediction is that eGFP levels can exhibit a transient peak during washout phase. This occurs because loading decreases before the regenerated protein is diluted below critical levels. This is evident in our experiments with high load levels (high aaRS input DNA concentrations), where a transient spike in eGFP expression occurred during washout before a decrease was observed (Fig. 2c–d, Supplementary Fig. 6).

To approximate the optimal DNA input concentrations for self-regeneration, we estimated aaRS protein synthesis rates for different DNA concentrations by using the ratio of aaRS to eGFP DNA, while assuming the same synthesis rate for all proteins, and comparing them with the estimated synthesis rate required to reach the minimum concentration needed for each aaRS (Supplementary Fig. 7a, Supplementary Table 4). In agreement with the observed data, we estimated 0.1 and 0.05 nM of DNA for AsnRS and LeuRS, respectively. Moreover, we confirmed these estimates based on the drop in eGFP synthesis rate for different DNA input concentrations (Supplementary Fig. 7b).

**T7 RNAP regeneration.** After testing two proteins essential for translation, we tested self-regeneration of an essential protein for transcription (Fig. 3a). For transcription the PURE system utilizes T7 RNAP, a single 99 kDa protein. As before, we carried out batch experiments to validate T7 RNAP synthesis in the PURE system (Supplementary Fig. 3), and essentiality of T7 RNAP (Fig. 3b). T7 RNAP could be successfully regenerated in the system and we carried out extensive DNA template titrations with concentrations varying over three orders of magnitude (Fig. 3c, Supplementary Fig. 8). By omitting the washout phase and extending the self-regeneration phase to 26 hours, we showed that T7 RNAP can be regenerated at steady-state for over 25 hours with a DNA input concentration of 0.5 nM (Fig. 3d).

To summarize the DNA titration results, we plotted eGFP expression levels as a function of T7 RNAP DNA template concentration at 11 hours of regeneration (corresponding to 15 hours after the start of the experiment), normalized to the

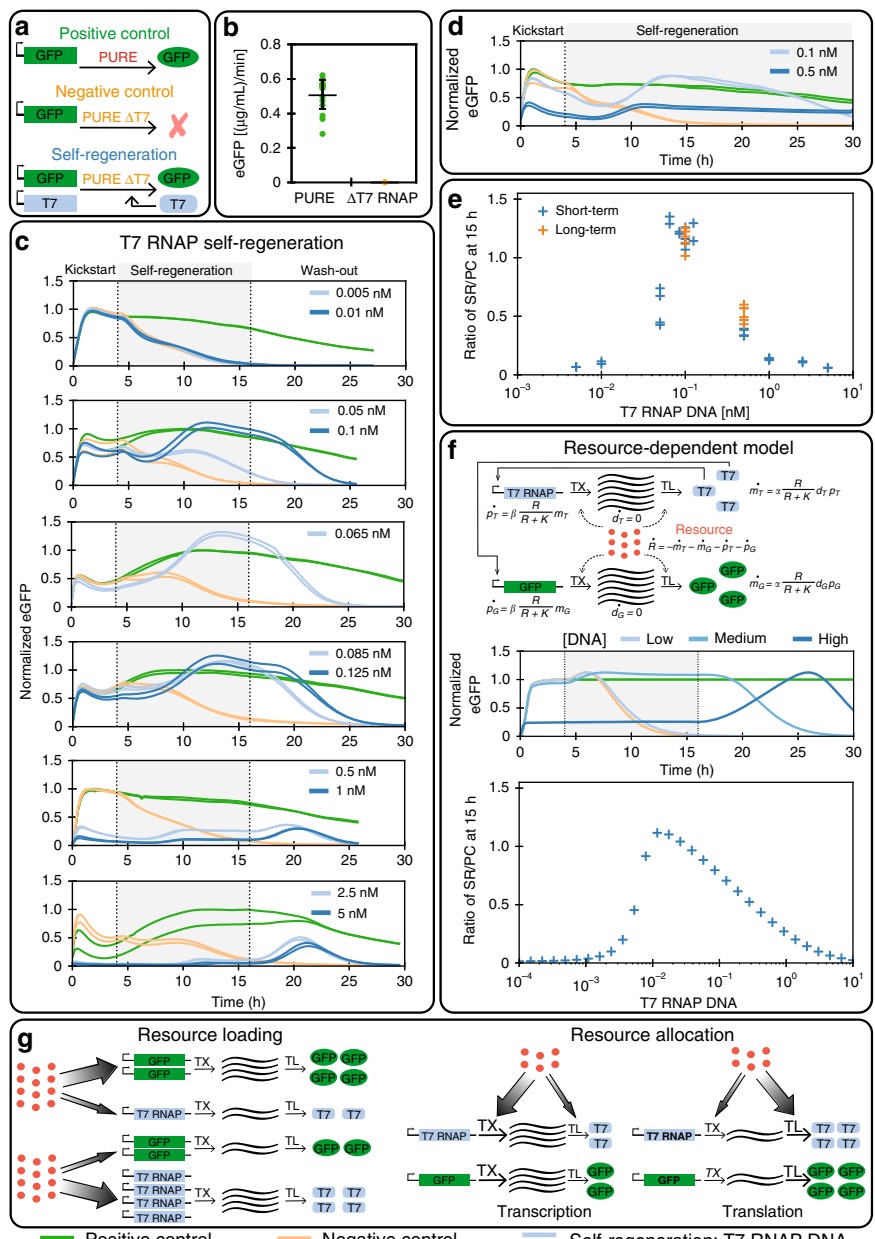

**Fig. 3 T7 RNAP regeneration. a** Overview of the T7 RNAP regeneration experiment. **b** eGFP batch synthesis rates for the full PURE and T7 RNAP ΔPURE systems (mean ± s.d. for PURE system (green, $n = 16$ technical replicates), and mean for ΔPURE systems (yellow, $n = 2$ technical replicates)). **c** T7 RNAP regeneration at different DNA template concentrations. **d** Long-term regeneration experiment: the self-regeneration phase was extended by omitting the washout phase. The results for all DNA concentrations tested and the appropriate mScarlet traces can be found in Supplementary Fig. 8. The level of eGFP is normalized to the maximum level attained in the positive control experiments (positive control: green, negative control: yellow, self-regeneration: blue). The composition of the PURE system used for the self-regeneration experiments is given in Supplementary Table 3, 2 nM of eGFP template was used for all experiments, T7 RNAP DNA template concentrations are indicated. **e** Ratio of eGFP levels of the self-regeneration experiments and the positive control at 15 hours as a function of T7 RNAP DNA template concentration. Each data point represents a single measurement. **f** Our resource-dependent model consists of seven ODEs and three parameters. DNA, mRNA, and protein concentrations are denoted by $d$, $m$, and $p$, and the subscripts $T$ and $G$ refer to T7 RNAP and eGFP, respectively. Simulation of a self-regeneration experiment: the switch between stages occurs at 4 and 16 hours. DNA for T7 RNAP was present at three qualitatively different concentrations, indicated as "low", "medium", and "high". All concentrations are non-dimensional. The level of GFP is normalized by the maximum level attained in the positive control experiment. The negative control corresponds to $d_T = 0$. **g** Schematic description of the concepts of resource loading and resource allocation. Resource loading is the distribution of a limited resource between two genes. Resource allocation is the distribution of a limited resource (red) between transcription (TX) and translation (TL). A detailed description of the different concepts can be found in Supplementary Fig. 19. Source data are available in the Source Data file.

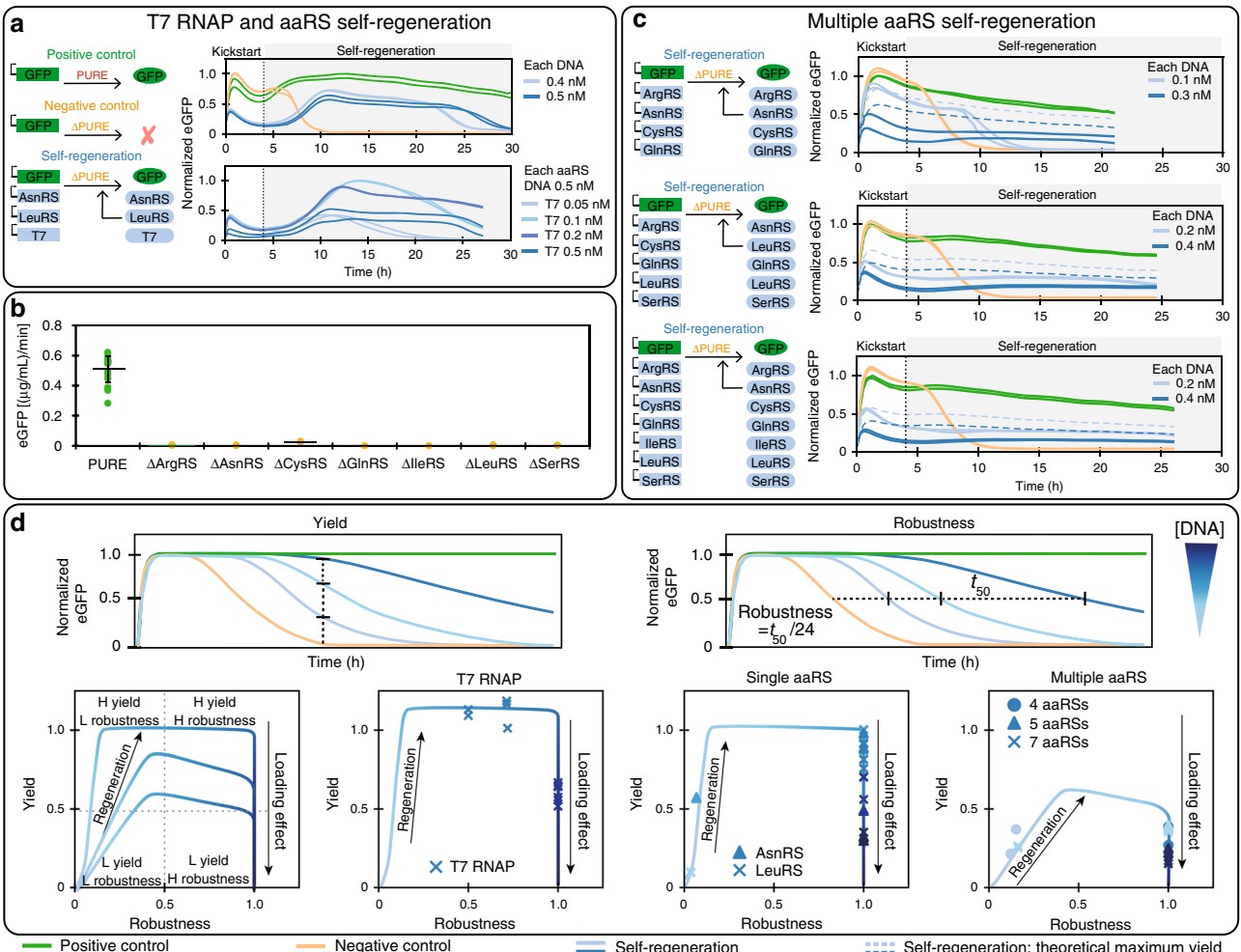

**Fig. 4 Multi-component regeneration. a** Combined T7 RNAP, AsnRS, and LeuRS regeneration. Overview of the experiments on the left, with experimental results shown on the right. The top graph shows results for all DNA templates at concentrations of 0.4 or 0.5 nM. The bottom graph shows results for a titration of T7 RNAP DNA template with both aaRS DNA templates held constant at 0.5 nM. **b** eGFP batch synthesis rates for the PURE system and ΔPURE systems lacking additional aaRSs (mean ± s.d. for PURE system (green, $n = 16$ technical replicates), and mean for ΔPURE systems (yellow, $n = 2$ technical replicates)). **c** Simultaneous regeneration of 4, 5, and 7 aaRSs. Overview of self-regeneration experiments on the left, with experimental results shown on the right. Results for all DNA concentrations tested and the corresponding mScarlet traces can be found in Supplementary Figs. 20, 21. All eGFP traces were normalized to the maximum eGFP fluorescence output in the positive control, with exception of the T7 RNAP titration in **a** for which eGFP traces were normalized to the maximum eGFP fluorescence (positive control: green, negative control: yellow, self-regeneration: blue). Dashed lines represent a theoretical maximal yield for the corresponding aaRS DNA loads, calculated from the positive control. PURE system compositions are given in Supplementary Table 3. In all, 2 nM of eGFP DNA template was used in all experiments, other DNA template concentrations are indicated. **d** Schematic description of the definition of yield and robustness. Using these two terms we plotted theoretical curves for the relationship between DNA input concentration, yield, and robustness and superposed experimental values obtained from T7 RNAP, single aaRS, and multiple aaRS self-regeneration experiments. All three systems follow a similar trajectory and can be described in terms of Pareto optimality. Source data is available in the Source Data file.

positive control expression levels (Fig. 3e). For lower DNA concentrations (<0.05 nM) we observe little or no eGFP expression, which we attribute to insufficient synthesis of T7 RNAP, similar to the results obtained for aaRSs. For high T7 RNAP DNA template concentrations (≥1 nM) resource competition similar to what was observed for AsnRS and LeuRS was taking place. This is also supported by the observed peak during the washout phase for high-input DNA template concentrations. Near optimal system performance within 80% of the control reaction occurred in a narrow DNA template concentration range of 0.65 nM to 0.125 nM. The curve is asymmetric, with higher sensitivity to low concentrations than to higher concentrations, providing insights into how system robustness can be engineered. Surprisingly, and unlike the aaRS experiments, we observed an

expression maximum that rises to a level of 1.3 above the positive control reactions, indicating that a simple resource competition model cannot account for the observed behavior.

To investigate whether our hypothesis of resource competition could be extended to explain the T7 RNAP observations, we created a minimal model of the transcription–translation system. Although transcription–translation systems can be described at varying levels of granularity, e.g., ref. [42–45], we chose to model the processes at the most coarse-grained level using coupled ordinary differential equations (ODEs) (Fig. 3f, Supplementary Fig. 9–18, Supplementary Table 5).

The model consists of transcription and translation of eGFP and T7 RNAP, which consumes a single resource species $R$. This species is a lumped representation of CTP, UTP, ATP, GTP, and

aminoacyl-tRNAs, which are consumed during transcription and/or translation. We model the transcription rate by a parameter $\alpha$, linearly dependent on DNA and T7 RNAP concentration, and modulated by the availability of resources using a Hill function $R/(R + K)$. Likewise, translation proceeds at a rate $\beta$, is linearly dependent on mRNA concentration, and is modulated by the same Hill function. The rate of consumption of $R$ is equal to the summed transcription and translation rates. The complete model consists of seven ODEs and three parameters, and is solved between discrete dilution steps to simulate chemostat operation.

This minimal model successfully captures the observed qualitative behavior including: (1) eGFP washout at low T7 RNAP DNA concentrations ($d_T$) in the self-regeneration phase, (2) low eGFP production followed by a peak in the washout stage at high $d_T$, and importantly (3) eGFP production in excess of the positive control at medium $d_T$ in the self-regeneration phase (Fig. 3f). At low $d_T$, mRNA concentration is low, whereas resources are abundant; translation rate is thus limited by mRNA concentration. High $d_T$ leads to increased resource consumption, so despite the presence of large amounts of mRNA, translation is limited by resources. Further analysis reveals that at intermediate $d_T$ concentrations, eGFP production can increase above the positive control during the self-regeneration phase. The model predicts that this is owing to a reallocation of resources from transcription to translation, once self-regeneration of T7 RNAP begins. This effect requires a resource-limited condition (Fig. 3g, Supplementary Fig. 19).

We developed an alternative resource-independent model, which only takes into account translational loading through a shared translational enzyme, which can also capture the observed optimum in the SR/PC ratio (Supplementary Fig. 16). In this case, the optimum is owing to a trade-off between mRNA concentration and enzyme availability. However it fails to predict the increase in eGFP production above the positive control during self-regeneration.

The modeling studies indicate that the requirement for an optimum in the SR/PC ratio is a coupling between eGFP and T7 RNAP expression, whether through a shared resource or a shared enzyme. However, the increase of eGFP above the positive control during the self-regeneration phase requires a resource-limited condition, and resource reallocation from transcription to translation (Fig. 3g, Supplementary Fig. 19). Although both models can be combined, or extended to incorporate more realistic effects, such as saturation of transcription rates with substrate concentration, time delays in the various processes, and more intricate mechanisms of resource usage, none of these are required to explain our observations, apart from the essential feature of gene expression coupled through a shared resource.

**Regeneration of multiple components**. Having demonstrated that proteins essential for translation or transcription could be regenerated individually, we explored whether multiple proteins could be regenerated simultaneously. We first tested if T7 RNAP, AsnRS, and LeuRS could be regenerated together. Initial DNA concentrations tested were 1× and 2× the minimal DNA concentrations which led to successful self-regeneration of individual proteins, but these concentrations were not sufficient for sustained self-regeneration of multiple proteins (Supplementary Fig. 20). Increasing DNA concentrations and maintaining 1:1 DNA template concentration ratios ultimately led to successful regeneration lasting 20–25 hours (Fig. 4a, Supplementary Fig. 20). Despite successfully regenerating for many hours, protein synthesis ultimately ceased under these conditions. Based on the T7 RNAP results and our computational modeling we hypothesized that a more optimal DNA ratio between T7 RNAP and the

aaRSs needed to be established, as we previously observed strong resource loading effects by T7 RNAP and an apparent insensitivity of optimal T7 RNAP DNA concentration in respect to overall loading. Consequently, we decided to retain a relatively high DNA concentration of 0.5 nM for both aaRSs, and titrated T7 RNAP DNA template (Fig. 4a). This had the desired effect and resulted in sustained regeneration at a T7 RNAP DNA concentration of 0.2 nM.

To explore the limits of the PURE transcription–translation system for self-regeneration, we tested whether several aaRSs could be regenerated simultaneously. We first carried out batch experiments to ensure efficient expression of the chosen aaRSs (Supplementary Fig. 3), as well as lack of expression if a given aaRS was omitted from the PURE system (Fig. 4b). As for the single aaRS experiments, we adjusted the concentrations of the various aaRS proteins (Supplementary Table 3) in the PURE system to ensure efficient washout. We gradually increased the number of aaRSs being regenerated from four to seven (Fig. 4c, Supplementary Fig. 21).

Based on eGFP synthesis rate and DNA ratios for the different conditions, we estimated that DNA inputs above 0.2 nM would be required for successful regeneration (Supplementary Fig. 22a). This was in agreement with the observed data and decreases in eGFP synthesis rate owing to loading (Supplementary Fig. 22b). We observed successful self-regeneration of up to 22 h for experiments with input DNA concentrations of 0.2 nM or above. DNA concentrations of 0.1 nM on the other hand led to rapid cessation of protein synthesis activity 10 h into the experiment. Furthermore, when DNA input concentrations of 0.2 nM were used we saw variations in the length of self-regeneration among experiments (Supplementary Fig. 21). The estimated synthesis levels are much higher than the concentrations of most aaRS diluted out of the reactor each cycle, with exception of ArgRS, where 0.027 (μg/mL)/min is diluted out, suggesting that optimization of DNA input for individual aaRSs could allow for better resource allocation and higher robustness.

eGFP levels are low when compared to the positive control. The positive control represents the maximum achievable eGFP steady-state levels in an otherwise unloaded system. Expression of 4–7 additional aaRS presents a considerable load on the system. When taking this load into account, self-regeneration of 4–7 aaRSs in addition to expressing eGFP reaches ~50% of the theoretically achievable yield (Fig. 4c), indicating that the total synthesis capacity of the system remained quite high.

These experimental results suggest that achieving successful self-regeneration depends on an interplay of several factors. To more quantitatively describe the system we defined the terms yield and robustness (Fig. 4d). We define yield as the level of non-essential protein such as eGFP that the system can synthesize during self-regeneration. In the case where an essential protein, for instance an aaRS, is missing, yield is zero. Expressing the aaRS will increase the yield, up to a point where the system's resources are preferentially directed towards aaRS production. At that point system yield begins to decrease again owing to loading. A second important parameter is system robustness. We consider a robust system to be able to sustain self-regeneration for at least 24 h. A non-robust system may temporarily reach steady-state self-regeneration, but changes in synthesis rates, DNA concentrations, or environmental conditions, can cause it to cease functioning. We therefore define robustness as the time the system self-regenerates beyond the negative control, normalized by 24 h. A system that self-regenerates for 24 h or longer receives a robustness score of 1, whereas systems that cease regeneration before 24 h receive a score between between 0 and 1.

Given these two parameters: yield and robustness, one can now describe the system in terms of Pareto optimality and determine

whether there exists a trade-off between yield and robustness. In Fig. 4d, we show the calculated values of yield against robustness for our experimental observations, as well as the theoretically expected relationship of yield, robustness, and DNA concentration. For an essential protein, increasing its expression constrains the yield of the system onto a Pareto front[46]. This is because low expression of that protein leads to large increases in yield as the protein begins to confer its advantage on the system. Above a critical concentration, the system is able to continuously regenerate, corresponding to a robustness of 1. Expressing the protein at higher levels than the critical value incurs a cost on the system, which is exhibited by decreasing yield due to loading.

For single protein self-regeneration it is indeed possible to reach maximal yield and robustness. However, whether that situation can be attained or not depends on the activity of the essential protein. Proteins with low activity require higher concentrations, and hence more resources, to produce. This thus limits the attainable yield of the system, and shifts the yield-robustness curve downwards. In severe cases (which we do not observe), yield may never reach 1. Regeneration of multiple proteins falls into this category: when several essential proteins are being regenerated, the available capacity to express other proteins becomes less. Nonetheless, it is possible to attain high robustness with a corresponding trade-off in yield. And finally, the range of DNA concentrations that give rise to high yield and high robustness are often quite narrow indicating that feedback regulation may become a necessary design requirement[47].

## Discussion

We demonstrate how a biochemical constructor could be created by implementing a transcription–translation system running at steady-state on a micro-chemostat that supplies the reaction with resources and energy. We showed that the system is capable of self-regenerating components of its core constructor by synthesizing proteins required for transcription and translation. We regenerated up to seven components simultaneously and show that system optimality is surprisingly similar to fitness landscapes observed in living systems[48], requires both minimizing resource loading and optimizing resource allocation, and can be described in terms of Pareto optimality.

Just like the universal constructor envisioned by von Neumann ~80 years ago, a biochemical universal constructor will consist of three components: (i) an instruction set (DNA), (ii) a core constructor (RNA and proteins), (iii) and a copy machine (proteins). The core constructor consists of RNAs and proteins that read and implement the information contained in the instruction set. The core constructor is capable of constructing copies of itself and of the copy machine. The copy machine consists of the protein components necessary for DNA replication, which copy the instruction set[14]. Similar to von Neumann's universal constructor, the biochemical constructor requires supply of resources and energy, which is also a necessary requirement for all living systems.

Although we show that creation of a biochemical constructor is feasible, a number of considerable challenges remain. It will be critical to develop a transcription–translation system with a high-enough synthesis rate to self-regenerate all of its components. The PURE system is currently orders of magnitude away from this target. We estimate that ~50% of all PURE proteins could be regenerated by the current PURE system, and that the total synthesis rate required is 25-fold above the current rate (Supplementary Fig. 23). These estimates do not yet include ribosome or tRNA synthesis. Current approaches to optimizing transcription–translation systems mainly focus on increasing component concentrations or adding components to the system,

which can give rise to overall higher synthesis yields but consequently also require higher synthesis rates to achieve self-regeneration[27,28,33]. Instead, optimizing protein synthesis rates and the ratio of protein synthesis rate to total amount of protein contained in the system will be important for development of a biochemical universal constructor. A second major challenge lies in achieving functional in vitro ribosome biogenesis[22,24,49]. The most promising near-term goal will be demonstration of steady-state self-replication of DNA. Several promising advances have recently been demonstrated in this area[14,15], although in vitro DNA replication efficiency likely needs to be improved in order to reach sustained steady-state DNA replication.

Achieving high yield and robustness will be as well important for the development of a universal biochemical constructor. These concepts are tightly connected to resource usage and loading effects recently described in cell-free systems[41] and living cells[50]. We showed that several components could be regenerated at the same time. However, finding optimal DNA concentrations for several components is critical to achieving sustained regeneration without unnecessarily loading the system. Moreover, our results and corresponding modeling suggest that specific components might have to be tightly regulated, and could benefit from active feedback regulation[47], especially once system complexity increases. Currently, self-regenerating systems can be optimized by varying individual DNA input concentrations in order to adjust protein synthesis rates for each component being regenerated. In the future, all genes will be encoded on a single "genome"[15,51], requiring expression strengths to be tuned by the use of synthetic transcription factors[52], promoters[53], terminators[54], and ribosome binding sites[55]. Work on a biochemical universal constructor thus provides ample challenges and opportunities for synthetic biology in the areas of protein biochemistry, tRNA synthesis, ribosome biogenesis, metabolism, regulatory systems, genome design, and system engineering.

The development of a universal biochemical constructor and the creation of synthetic life are exciting prospects and recent progress in technology and biochemistry are making these seemingly plausible goals. Many challenges remain, but pieces to the puzzle are being added at an increasing rate. It is thus not far-fetched to consider that synthetic life, engineered by humans from basic building blocks, may be a possibility.

## Methods

**Materials.** E. coli BL21(DE3) and M15 strains were used for protein expression. E. coli RB1 strain[56] originally obtained from G. Church (Wyss Institute, Harvard University, USA) was used for His-tag ribosome purification. All plasmids encoding PURE proteins used in this work were originally obtained from Y. Shimizu (RIKEN Quantitative Biology Center, Japan). Plasmid encoding mScarlet was a gift from P. Freemont (Imperial College London, UK).

Linear template DNA for in vitro eGFP synthesis (Supplementary Table 6) was initially prepared by extension PCR from a pKT127 plasmid as described previously[35] and cloned into a pSBlue-1 plasmid. The DNA fragment used for PURE system characterization and self-replication experiments was amplified from this plasmid by PCR. Linear DNA fragments encoding different proteins used for self-regeneration experiments were prepared by extension PCR from their respective plasmids. Primer sequences are listed in Supplementary Table 7. All DNA fragments were purified using DNA Clean and Concentrator-25 (Zymo Research). DNA was eluted in nuclease-free water instead of elution buffer, and its concentration was quantified by absorbance (NanoDrop, ThermoFisher). Double stranded Chi DNA[57] was prepared by annealing two primers listed in Supplementary Table 6.

**Ribosome purification.** All buffers used in this work are listed in Supplementary Table 8. All buffers were filtered (Flow Bottle Top Filters, 0.45 μm aPES membrane) and stored at 4 °C. 2-mercaptoethanol was added immediately before use. Ribosomes were prepared from E. coli RB1 strain by His-tag purification[56]. E. coli RB1 strain was grown overnight in 3 mL LB media at 37 °C. In all, 4 × 3 mL of the overnight culture was used to inoculate 4 × 500 mL of LB in a 1 L baffled flask. Cells were grown at 37 °C, 260 RPM to exponential phase (3–4 h), pooled together and harvested by centrifugation (3220 RCF, 20 min, at 4 °C), and stored at −80 °C. The

cells were then resuspended in 15 mL suspension buffer and lysed by sonication on ice (Vibra cell 75186, probe tip diameter: 6 mm, 11 × 20 s:20 s pulse, 70% amplitude). Cell debris was removed by centrifugation (21130 RCF, 20 min, at 4 °C). The recovered fraction was filtered with a GD/X syringe filter membrane (0.45 mm, polyvinylidene difluoride, Whatman).

Ribosomes were purified using 5 mL IMAC Sepharose 6 FF (GE Healthcare) by Ni-NTA gravity-flow chromatography. The corresponding buffers were prepared by mixing buffer C and buffer D at the required ratios. After the column was equilibrated with 30 mL of lysis buffer (100% buffer C), the prepared lysate solution was loaded onto the column. The column was washed with 30 mL of lysis buffer (100% buffer C), followed by 30 mL of wash buffer 1 (5 mM imidazole), 60 mL of wash buffer 2 (25 mM imidazole), 30 mL wash buffer 3 (40 mM imidazole), 30 mL wash buffer 4 (60 mM imidazole) and eluted with 7.5 mL elution buffer (150 mM imidazole). Ribosomes from two purifications were pooled together (~15 mL) and subjected to buffer exchange using a 15 mL Amicon Ultra filter unit with a 3 kDa molecular weight cutoff (Merck). All centrifugation steps were performed at 3220 RCF and 4 °C. The elution fraction was concentrated to 1 mL (60 min). The concentrated sample was then diluted with 15 mL of ribosome buffer and re-concentrated to 1 mL (60–70 min); this step was repeated three times. The recovered ribosomes (1 mL) were further concentrated using a 0.5 mL Amicon Ultra filter unit with a 3 kDa molecular weight cutoff (Merck) by centrifugation (14,000 RCF, at 4 °C). Ribosome concentration was determined by measuring absorbance at 260 nm of a 1:100 dilution. An absorbance of 10 for the diluted solution corresponds to a 23 μM concentration of undiluted ribosome solution. Final ribosome solution used for in vitro protein synthesis was prepared by diluting to 3.45 μM. The usual yield is ~0.75 mL of 3.45 μM ribosome solution.

**PURE system preparation**. All buffers used in this work are listed in Supplementary Table 8. All buffers were filtered (Flow Bottle Top Filters, 0.45 μm aPES membrane) and stored at 4 °C. 2-mercaptoethanol was added immediately before use. Proteins were purified by Ni-NTA gravity-flow chromatography as described previously[26]. All cultures were grown at 37 °C, 250 rpm. Overnight cultures were grown in 3 mL of LB supplemented with 100 μg/mL of ampicillin and/or 50 μg/mL of kanamycin. Each strain was inoculated in a flask with 2 L of LB. Cells were grown 2 h before induction with 0.1 mM of isopropyl β-D-1-thiogalactopyranoside (IPTG) for 3 h, then harvested by centrifugation and stored at −80 °C. The cells were resuspended in 30 mL of buffer A and lysed by sonication on ice (Vibra cell 75186; probe tip diameter: 6 mm; 8 × 20 s:20 s pulse; 70% amplitude). Cell debris was removed by centrifugation (25,000 RCF, 20 min, 4 °C). The supernatant was mixed with 2–3 mL of equilibrated resin (described below), and incubated for up to 2 h, at 4 °C. After the incubation, lysate was allowed to flow through the column. The column was washed with 30 mL of a wash buffer (95% buffer A, 5% buffer B) and eluted with 15 mL of an elution buffer (10% buffer A, 90% buffer B). The elution fraction was dialyzed against HT buffer (2×) and stock buffer and stored at −80 °C. Protein concentrations were estimated by absorbance at 280 nm and calculated protein extinction coefficients. When a higher protein concentration was required, the protein solution was concentrated using a 0.5 mL Amicon Ultra filter unit (Merck). Different PURE protein formulations are summarized in Supplementary Table 3. Different PURE or ΔPURE systems were prepared by supplying the corresponding ΔPURE systems with the omitted protein or buffer solution, respectively.

**Ni-NTA resin preparation and regeneration for ribosome purification**. In all, 5 mL IMAC Sepharose 6 FF (GE Healthcare) was pipetted into Econo-Pac chromatography columns (Bio-Rad), and charged with 15 mL of 100 mM nickel sulfate solution. The charged column was washed with 50 mL of demineralized water. After protein purification, columns were regenerated with 10 mL of buffer containing 0.2 M EDTA and 0.5 M NaCl, and washed with 30 mL of 0.5 M NaCl, followed by 30 mL of demineralized water, and stored in 20% ethanol at 4 °C.

**Energy solution preparation**. Energy solution was prepared as described previously[26] with slight modifications. In all, 2.5× energy solution contained 0.75 mM of each amino acid, 29.5 mM magnesium acetate, 250 mM potassium glutamate, 5 mM ATP and GTP, 2.5 mM CTP, UTP, and TCEP (tris(2-carboxyethyl)phosphine hydrochloride), 130 $U_{A260}$/mL tRNA, 50 mM creatine phosphate, 0.05 mM folinic acid, 5 mM spermidine, and 125 mM HEPES (4-(2-hydroxyethyl)-1-piperazineethanesulfonic acid).

**Batch in vitro protein expression experiments**. Batch PURE reactions (5 μL) were established by mixing 2 μL of 2.5× energy solution, 0.9 μL of 3.45 μM ribosomes (final concentration: 0.6 μM), 0.65 μL of PURE proteins (Supplementary Table 3), DNA template, and brought to a final volume of 5 μL with addition of water. All reactions measuring eGFP expression were prepared as described above with eGFP linear template at a final concentration of 4 nM and incubated at 37 °C at constant shaking for 3 h, and measured (excitation: 488 nm, emission: 507 nm) on a SynergyMX platereader (BioTek). The eGFP production rate was calculated between 20 and 50 min based on an eGFP calibration curve (Supplementary Fig. 24a). Reactions expressing other proteins were prepared as described above and supplemented with 0.2 μL FluoroTect GreenLys (Promega). DNA templates

were used at a final concentration of 2 nM and the reactions were incubated at 37 °C for 3 h.

**SDS-PAGE gels**. PURE reactions (5 μL) labeled with FluoroTect GreenLys (Promega) were incubated with 0.8 μg or 0.2 μL of RNAse A solution (Promega) and incubated for 30 min at 37 °C and subsequently analyzed by SDS-PAGE using 10-well 4–20% Mini-PROTEAN TGX Precast Protein Gels (Bio-Rad). Gels were scanned (AlexaFluor 488 settings, excitation: Spectra blue 470 nm, emission: F-535 Y2 filter) with a Fusion FX7 Imaging System (Vilber) and analyzed with ImageJ. Protein sizes were calculated based on a BenchMark^TM Fluorescent Protein Standard (Invitrogen).

**Fabrication and design of the microfluidic device**. The microfluidic device was fabricated by standard multilayer soft lithography[58]. Detailed device preparation, operation, and characterization are described previously[37]. The device with eight reactors and nine fluid inputs (Fig. 1b) is based on a previous design[35]. Molds for the control and the flow layer were fabricated on separate wafers by standard photolithography techniques and patterned with photoresist to produce channels with the heights stated (Supplementary Fig. 1c). For the control layer, a silicon wafer was primed in an oxygen plasma processor for 7 minutes (TePla 300), and SU-8 photoresist (GM 1070, Gersteltec Sarl) was spin-coated onto the wafer yielding a height of 40 μm. After relaxation and soft bake, the wafer was illuminated using a chrome mask for 18.2 s (365 nm illumination, 20 mW/cm² light intensity) on a Süss MJB4 mask aligner, followed by a post-exposure bake, the wafer development with propylene glycol monomethyl ether acetate and a hard bake. For the flow layer, a silicon wafer treated with HMDS (hexamethyldisilazane) vapor (YesIII primer oven) was spin-coated with AZ 9260 photoresist (Microchemicals GmbH) to a height of 15 μm. After baking and relaxation time, the coated wafer was exposed two times 18 s with a 10 s wait period between (20 mW/cm² light intensity) on a Süss MJB4 mask aligner. The wafer was developed with AZ 400K developer and baked at 175 °C for 2 hours.

The microfluidic chips were fabricated from PDMS by standard multilayer soft lithography. Each of the wafers was treated with TMCS (trimethylchlorosilane). For the flow layer PDMS with an elastomer to crosslinker ratio of 5:1 was prepared and poured over the wafer and place to a desiccator for 40 min. For the control layer, PDMS with a 20:1 elastomer to crosslinker ratio was spin coated at 1400 rpm on to the wafer, and left to sit for 40 min before baking. Both PDMS coated wafers were baked in the oven at 80 °C for 20 minutes. After the layers were aligned by hand and the aligned devices were placed in the oven for 90 minutes. The bonded chip was punched using a 900 mm pin and plasma bonded to a glass slide.

**Device setup**. To prime the chip, control lines were filled with phosphate-buffered saline (PBS) and pressurized at 1.38 bar. The flow layer was primed with a solution of 2% bovine serum albumin in 0.5× PBS. For washes between loading steps, 10 mM TRIS buffer (pH = 8) was used. For the experiments energy, PURE, and DNA solutions were mixed in the microfluidic reactors on the microfluidic chip in a 2:2:1 ratio. The peristaltic pump was actuated at 20 Hz to mix the solutions. Every 15 min, the reactor was imaged and a 20% fraction of the reactor volume was replaced with fresh components with the same 2:2:1 ratio. Details on the operation of the microfluidic chip can be found in Supplementary Tables 1 and 2. In all, 2.5× energy solution was prepared as described above. In all, 2.5× PURE or ΔPURE solutions were prepared by mixing the desired protein solutions (Supplementary Table 3) with ribosomes (final concentration: 0.6 μM) and supplied with 10 μM TCEP (final concentration: 4 μM). The PURE solution was supplemented with mScarlet protein to allow visualization, and the solutions were brought to final volume with the addition of water. The DNA solution at five times its final concentration was prepared by mixing the desired linear templates and Chi DNA. The final concentration of eGFP reporter in the reaction was 2 nM, the Chi DNA was used at a final concentration of 1.25 μM. The Chi decoys were added to help mitigate potential DNA absorption and degradation, whereas the DNA solution is stored in the FEP tubing before it is added to the chip.

**Data acquisition and analysis**. Solenoid valves, microscope, and camera were controlled by a custom Matlab and LabVIEW program. The chip and microscope stage were enclosed in an environmental chamber at 34 °C. Green and red fluorescence was monitored over time on an automated inverted fluorescence microscope (Nikon), using 20× magnification and FITC/mCherry filters. The microscope hardware details are described in ref. [37].

The fluorescence images were analyzed and corrected in Python, by subtracting the background fluorescence of a position next to the fluidic channel. The fluorescence signal was normalized in respect to maximal positive control signal intensity in a given experiment, or to the overall maximal intensity if a positive control was not included. The eGFP synthesis rate was calculated based on an eGFP calibration curve (Supplementary Fig. 24b) and dilution rate.

**Reporting summary**. Further information on research design is available in the Nature Research Reporting Summary linked to this article.

## Data availability

The authors declare that all data supporting the findings of this study are available within the article and its supplementary information files or from the corresponding author upon reasonable request. Source data are provided with this paper.

## Software and code availability

The data analysis in this study was performed with Python code and in Numbers 10.1, microfluidic control and data acquisition code were performed by Labview code, which will be made available from the corresponding author upon reasonable request. Supporting code for Fig. 3f is available on GitHub at https://github.com/lbnc-epfl, https://doi.org/10.5281/zenodo.4160155[59]. Microfluidic design files are available at http://lbnc.epfl.ch/microfluidic_designs.html. Source data are provided with this paper.

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

## Acknowledgements

This work was supported by Human Frontier Science Program Grant RGP0032/2015; the European Research Council under the European Union's Horizon 2020 research and innovation program Grant 723106; and EPFL. N.L. is supported by a Chancellor's Fellowship from the University of Edinburgh.

## Author contributions

B.L. performed experiments. N.L. performed the computational analysis. B.L., N.L., and S.J.M. designed experiments, analyzed data, and wrote the manuscript.

## Competing interests

The authors declare no competing interest.
