## [Peer Review File · Nature Communications]

Reviewers' Comments:

Reviewer #1:

Remarks to the Author:

Report for Maerkl paper.

In this manuscript, Maerkl and his group constructed a system of self-regenerating cell-free protein expression based on the micro fluidics device. The system was designed to carry out three phases of reaction (i.e. kickstart, self-regeneration, and washout) to observe long time protein synthesis reaction using eGFP. Self-reproduction is one of the biggest issues in the research of creation of cellular life by assembling molecules and genes in the field of synthetic biology. The challenge of how to implement self-reproduction phenomenon in artificial (synthetic) cell is classified in two aspects. One is self-reproduction of outer envelope of the cell and another one is that of the internal molecules and genetic information. Cell-free system synthesizing cell-free component proteins is, therefore, important issue for the realization of creation of living cell, and, the reviewer believes, Maerkl's system worked well in this point.

The main finding of this research is that minimizing resource competition and optimizing resource reallocation are important for achieving robust self-reproducing cell-free system. This claim has been well supported by provided evidences and models. Additionally, the authors claim that the amount of input template DNA and ratio between each gene are also important and its optimal range is very narrow. Although the manuscript is well described and, basically, there is no problem for the publication, the reviewer has some points that cannot be understand accurately. One is about the expression maximum arose more than the positive control when T7 RNAP was regenerated (Fig. 3C and D). The authors explained this was happened because the system was shifted from the simple resource competition mode to the resource reallocation mode. Fig. 3G cartoon is clearly showing the differences of two modes. However, what the authors performed in this experiment is just changing the amount of input gene of T7RNAP that co-expressed with eGFP. The reviewer somehow cannot fully understand why this mode shift was occurred in the exactly same cell-free system just by changing the amount of input DNA. Or, am I misunderstanding something?

Another one is about the regeneration of multiple components. The result of Fig. 2 and 3 showed that the optimal DNA concentration was tightly decided for the stable regeneration, for example optimum concentration of AsnRS is 0.1 nM and Leu is about 0.1 nM, and too much DNA shows negative effects. However, when five or more components are synthesized, much higher DNA concentration for each component gene is required (Fig. 4). Why so much high concentration was needed? In such condition I can image that the resource competition seems to be accelerated in the system.

The reviewer would like to requests authors to explain again what reaction dynamics are actually happening in the system regarding these points.

The follows are additional minor points that should be revised before publication.

Line 6 and 21, A report by Suetsugu et al. is also suitable for the citation of DNA replication. (<https://academic.oup.com/nar/article/45/20/11525/4209619>)

Line 21, 42 and 319, Recent reports by Shimizu et al is also suitable for the citation of in vitro ribosome reconstitution. (<https://www.nature.com/articles/s42003-020-0874-8>)

Figure 1, This figure is explaining overview of the developed micro fluidic (MF) system for the study of self-reproduction of cell-free system. However, it is somehow not very easy to image the actual MF device. In particular, Fig. 1B may not give a direct understanding of how the device works. The color of red for "Control" is too strong to follow the blue "Flow" lines. The illustrates on the right side of B could be also improved by some tricks. Perhaps, putting a photo of the device may help to image what is going on.

Figure 2C and D, The authors show the traces of PC, SR, and washout on the graph each by two lines. But these are somehow disturbing to follow the results. Maybe single line as an average of the results is OK. And more detailed graph can be post in Supporting Information. This is also true for the Figure 3 and 4.

Line 332, In this study, all the template DNAs are introduced as linear DNAs. However, in the future, toward the construction of self-reproducible artificial cell, all gene should be arranged on a single circular DNA as mini genome. In such condition, the author should explain how to regulate expression level of each gene that requires very tight expression range for achieving self-reproduction.

Reviewer #2:

Remarks to the Author:

The article by Lavickova and colleagues shows self-regeneration in a model synthetic cell system. The paper is well written, figures are clear and informative and experimental details are well assembled.

The title and the claims made in the paper are somewhat misleading. The stated self-regeneration is not synthesis of building blocks from cell cycle. The microfluidics encapsulated cell-free protein expression system is far from a "synthetic cell" model.

The Authors make a claim of "mapping genotype-phenotype landscapes", while in fact the correlation is only traced between couple genes and their protein products.

Minor issues:

The role of Chi DNA is not clearly explained. This is PURE system, so the typical function of Chi in blocking RecBCD nuclease is not necessary.

Could you please explain why Leucyl-tRNA synthetase and Asparaginyl-tRNA synthetase were picked? A rationale for choosing those two would help to understand the setup of the work.

Why was 34C picked as incubation temperature? As far as I know, optimal PURE incubation is 37C, and optimal for eGFP maturation is 30C. It would be helpful to give rationale for the temperature Authors used.

The Figure S1A is unclear, it is hard to understand how separated inputs become outputs on the microfluidic diagram.

For most time course graphs on figures 2, 3 and 4: the traces do not have error bars.

Could you, please, explain what does the "kickstart" phase of the reaction mean?

On many time course traces on figures 2, 3 and 4, there appears to be GFP signal coming up as soon as ~half an hour (I'm approximating in the time between 0 and 5h labeled on axis). This is extremely fast for a signal from GFP in PURE system. Maturation time of eGFP variant (Kremers 2007) is 128 minutes, and best new mutants I've seen mature as fast as little over an hour. Do Authors have a particularly fast maturing variant?

The reader is referred to protocols.io for the details of the construction of microfluidic rig and imaging system. As far as I know, protocols.io is not peer reviewed. In fact, even relying on unsecured community voting system to validate protocols, this protocol only has 1 endorsement (and that's from one of the authors).

I would suggest that such crucial part of the paper as preparation of novel hardware and data collection should be included in the methods for this manuscript, to be reviewed by a subject matter expert before publication.

The synthesis of T7 RNA polymerase from a gene encoded under T7 promoter was demonstrated before.

This is a very interesting part of the paper, and the resource allocation discussion is interesting and certainly could be expanded. This is, however, not an example of self-regeneration. Minor

point: half-life of T7RNAP activity would be useful to demonstrate at which time point the replacement of endogenously synthesized T7RNAP is complete.

In my opinion, there is technical novelty in this paper, and the problem of resource allocation in cell-free transcription and translation systems is definitely timely and worth studying. However, the paper uses "regeneration" systems that has been demonstrated before (T7 RNAP synthesis under T7 promoter), and the conclusions made in this paper are too general.

The authors study in great detail regeneration of few key elements of PURE system, but they do not provide rationale of how those studies can help build completely self-regenerating cell-free protein expression system, and consequently a synthetic minimal cell.

Overall, the paper is technically sound and represents great experimental system, worked out with impressive details. However, the conclusions made in the paper are not supported by the experimental data. Calling the system self-regenerating is an overstatement, and there is not enough of evidence to sufficiently back up a claim that this is a general model for self-regenerating synthetic cell.

We thank the editor and reviewers for their valuable time to assess our manuscript, and their helpful comments. We addressed all comments in a point-by-point response below and revised the manuscript accordingly. We note that both reviewers considered our work to be technically sound. Reviewer #1 recognized the value and importance of this work towards the development of a synthetic cell, and reviewer #2 appreciated the technical developments as well as our description of resource allocation in cell-free systems.

Reviewer #1 (Remarks to the Author):

Report for Maerkl paper.

In this manuscript, Maerkl and his group constructed a system of self-regenerating cell-free protein expression based on the micro fluidics device. The system was designed to carry out three phases of reaction (i.e. kickstart, self-regeneration, and washout) to observe long time protein synthesis reaction using eGFP. Self-reproduction is one of the biggest issues in the research of creation of cellular life by assembling molecules and genes in the field of synthetic biology. The challenge of how to implement self-reproduction phenomenon in artificial (synthetic) cell is classified in two aspects. One is self-reproduction of outer envelope of the cell and another one is that of the internal molecules and genetic information. Cell-free system synthesizing cell-free component proteins is, therefore, important issue for the realization of creation of living cell, and, the reviewer believes, Maerkl's system worked well in this point.

The main finding of this research is that minimalizing resource competition and optimizing resource reallocation are important for achieving robust self-reproducing cell-free system. This claim has been well supported by provided evidences and models. Additionally, the authors claim that the amount of input template DNA and ratio between each gene are also important and its optimal range is very narrow. Although the manuscript is well described and, basically, there is no problem for the publication, the reviewer has some points that cannot be understand accurately.

- One is about the expression maximum arose more than the positive control when T7 RNAP was regenerated (Fig. 3C and D). The authors explained this was happened because the system was shifted from the simple resource competition mode to the resource reallocation mode. Fig. 3G cartoon is clearly showing the differences of two modes. However, what the authors performed in this experiment is just changing the amount of input gene of T7RNAP that co-expressed with eGFP. The reviewer somehow cannot fully understand why this mode shift was occurred in the exactly same cell-free system just by changing the amount of input DNA. Or, am I misunderstanding something?
 - ⇒ We realized that the schematic in Figure 3G was not sufficient to explain this rather complex concept. We therefore revised the manuscript and included a new Supplementary Figure S19 describing this in more detail.
 - ⇒ The difference arises from changing T7 RNAP concentrations, due to expression and dilution of T7 RNAP during the experiment. Therefore, the system's concentrations change during the experiment (Fig. S19C). In the kickstart phase, T7 RNAP is both synthesized and supplied in the PURE system pushing the resources to transcription. In the self-regeneration phase, no T7 RNAP is provided in the Δ PURE leading to release of resources from transcription and an increase in GFP production. Likewise, the positive control contains exogenously added T7

RNAP. If this concentration is above the optimal value then less GFP will be produced.

- Another one is about the regeneration of multiple components. The result of Fig. 2 and 3 showed that the optimal DNA concentration was tightly decided for the stable regeneration, for example optimum concentration of AsnRS is 0.1 nM and Leu is about 0.1 nM, and too much DNA shows negative effects. However, when five or more components are synthesized, much higher DNA concentration for each component gene is required (Fig. 4). Why so much high concentration was needed? In such condition I can image that the resource competition seems to be accelerated in the system. The reviewer would like to requests authors to explain again what reaction dynamics are actually happening in the system regarding these points.
 - ⇒ PURE protein synthesis capacity saturates at a total DNA concentration of around 1 nM (Line 131, Fig. S5). As we are using DNA concentrations above 2 nM, we are working in a saturated system, where resource loading occurs (Figure 3G, S19A, B). If only a single aaRS is synthesized, less DNA will be required to synthesize the required protein amounts. By adding DNA for different aaRSs, we introduce higher DNA competition not only to eGFP synthesis but also to aaRSs synthesis. Due to this “competition”, we needed to add more DNA for each aaRSs to ensure sufficient synthesis rates. Finally, the required DNA concentrations were only ~2-4 times higher compared to the optimal concentration for a single aaRS DNA template, and we synthesized 4-7 times more aaRSs variants (1 aaRS versus 4-7 aaRSs being synthesized simultaneously).

The follows are additional minor points that should be revised before publication.

- Line 6 and 21, A report by Suetsugu et al. is also suitable for the citation of DNA replication. (<https://academic.oup.com/nar/article/45/20/11525/4209619>)
 - ⇒ We revised the manuscript accordingly.
- Line 21, 42 and 319, Recent reports by Shimizu et al is also suitable for the citation of in vitro ribosome reconstitution. (<https://www.nature.com/articles/s42003-020-0874-8>)
 - ⇒ We revised the manuscript accordingly.
- Figure 1, This figure is explaining overview of the developed micro fluidic (MF) system for the study of self-reproduction of cell-free system. However, it is somehow not very easy to image the actual MF devise. In particular, Fig. 1B may not give a direct understanding of how the device works. The color of red for “Control” is too strong to follow the blue “Flow” lines. The illustrates on the right side of B could be also improved by some tricks. Perhaps, putting a photo of the device may help to image what is going on.
 - ⇒ We revised the manuscript accordingly.
 - ⇒ We modified Figure S1, where we included images of the actual device as suggested by the reviewer and microscope images of the flow lines filled with a fluorescent dye. We also included a video, illustrating simple chemostat operations.

- ⇒ A detailed description of typical and essential experimental operations, and characterization of the chip are given in the now peer-reviewed cited protocol (Laohakunakorn, 2021, in press), and currently available as a preprint (Laohakunakorn, 2020)
- Figure 2C and D, The authors show the traces of PC, SR, and washout on the graph each by two lines. But these are somehow disturbing to follow the results. Maybe single line as an average of the results is OK. And more detailed graph can be post in Supporting Information. This is also true for the Figure 3 and 4.
 - ⇒ We value the reviewer suggestion to make our manuscript more readable. However, we keep the separate lines. We believe it is more informative to the reader than averaging the replicates, although it might not be the most esthetically pleasing. Plotting the lines separately clearly shows the spread among replicates, which is in most cases negligible. Showing the raw data traces also conforms with Nature’s policies on data presentation.
- Line 332, In this study, all the template DNAs are introduced as linear DNAs. However, in the future, toward the construction of self-reproducible artificial cell, all gene should be arranged on a single circular DNA as mini genome. In such condition, the author should explain how to regulate expression level of each gene that requires very tight expression range for achieving self-reproduction.
 - ⇒ We believe that we described exactly this topic in sufficient detail in the manuscript as it is very interesting to us as well. Any additional detail and more explicit discussion would be outside the scope of this manuscript.
 - ⇒ Page 19, Line 332: In the future, all genes will be encoded on a single ‘genome’ [15, 49], requiring expression strengths to be tuned by the use of synthetic transcription factors [50], promoters [51], terminators [52], and ribosome binding sites [53].

Reviewer #2 (Remarks to the Author):

The article by Lavickova and colleagues shows self-regeneration in a model synthetic cell system.

The paper is well written, figures are clear and informative and experimental details are well assembled.

- The title and the claims made in the paper are somewhat misleading. The stated self-regeneration is not synthesis of building blocks from cell cycle. The microfluidics encapsulated cell-free protein expression system is far from a “synthetic cell” model.
 - ⇒ We were very careful to not call our system a ‘synthetic cell’, and instead and deliberately chose to use the phrase a ‘synthetic cell model’. Attempts to build a synthetic cell involve many diverse examples, which typically focus on a single or few essential subsystems and the term synthetic cell has been used (in our opinion carelessly in many instances) to describe various approaches, aspects and concepts in the field ((van Nies, 2018), (Xu, 2010), (Bartelt, 2018), (Garamella, 2019), (Gopfrich, 2019), (Weisse, 2018)). In addition, the term synthetic cell is often used to refer to encapsulated cell-free systems with no additional or single sub-functionality (review: Eilenberger, 2019) and the term artificial cell is used in literature in relation to on-chip compartments “encapsulating” cell-free reactions ((Tayar, 2017), (Karzbrun, 2014), (Sato, 2019)). We believe our work is carried out in a similar spirit, where the essential

functionality of self-regeneration is demonstrated, and thus would even warrant the label ‘synthetic cell’. Especially given the rather mirky definition that this terms now has acquired due to the very broad use by other research groups. We do believe that the term synthetic cell should be used more sparingly and we thus decided to clearly differentiate this work from a bona fide synthetic cell by calling it a “synthetic cell model”.

- ⇒ We describe that we are working towards achieving self-replication of a minimal TX-TL system, which requires regeneration of proteins, ribosomes, tRNAs, and DNA (introduction, 2nd paragraph). This is a specific target which is consistent with the von Neumann constructor, and does not necessarily incorporate or require more complex biological features such as biosynthesis of building blocks or a coordinated cell cycle.
- ⇒ We defined the term self-regeneration and that we are focusing on regeneration of essential components. We disagree with the reviewer that synthesis of essential components in our case is not regeneration, as we do not only synthesize the components, but use them to maintain system activity over a prolonged period of time, showing that they are active and can support synthesis in the system; therefore they do regenerate themselves. Moreover, we were very careful to avoid using other common terms used in relation to "living" artificial cell, like replication, self-replication, self-reproduction which might be misleading, and used the term regeneration which was used with similar implications in Libicher, 2020.

- The Authors make a claim of “mapping genotype-phenotype landscapes”, while in fact the correlation is only traced between couple genes and their protein products.
 - ⇒ Our ‘genome’ consists of 36 non-ribosomal protein coding genes, and we measured 8/36 which is a sizeable fraction.
 - ⇒ But we do recognize that “mapping landscapes” might imply a much finer resolution, and we therefore changed this terminology to “quantitated genotype-phenotype relationships”

Minor issues:

- The role of Chi DNA is not clearly explained. This is PURE system, so the typical function of Chi in blocking RecBCD nuclease is not necessary.
 - ⇒ The reviewer is correct that Chi DNA is usually not required for use in the PURE system. However, as the linear DNA was stored for a prolonged time at 34°C, we added the Chi decoys to help prevent any potential absorption and degradation of our linear DNA templates while the solution is stored in the FEP tubing before it is added to the chip.
 - ⇒ We revised the manuscript accordingly: “The Chi decoys were added to help mitigate potential DNA absorption and degradation, while the DNA solution is stored in the FEP tubing before it is added to the chip.”
- Could you please explain why Leucyl-tRNA synthetase and Asparaginyl-tRNA synthetase were picked? A rationale for choosing those two would help to understand the setup of the work.
 - ⇒ aaRSs were chosen primarily because of their essentiality in the system (without these enzymes no protein synthesis occurs in the system).
 - ⇒ More precisely, the two aaRSs were chosen based on previous research (Awai 2015) as they are highly soluble and have a good ratio of aminoacylation activity to

translation in the PURE system. Moreover, they are also of very different size (AsnRS 52.6 kDa, IleRS 97.2 kDa), which is important to test when using in-vitro protein synthesis, as different size proteins might be expressed with different efficiencies.

- Why was 34C picked as incubation temperature? As far as I know, optimal PURE incubation is 37C, and optimal for eGFP maturation is 30C. It would be helpful to give rationale for the temperature Authors used.
 - ⇒ We addressed the choice of 34°C in the manuscript. The primary reason being that choosing a 34°C temperature decreased degradation of reagents prior to introduction into the chemostat (the reagents are stored in tubing off-chip, but are at the same temperature as the device, i.e. 34°C). Higher temperatures improve protein synthesis but decrease reagent stability, and lower temperatures decrease protein synthesis but increase reagent stability.
 - ⇒ See line 69 page 6: “Secondly, reaction temperature was set to 34°C, which decreased PURE degradation with only a minor decrease in protein synthesis rate (Supplementary Fig. S2).”
- The Figure S1A is unclear, it is hard to understand how separated inputs become outputs on the microfluidic diagram.
 - ⇒ We revised the manuscript accordingly.
 - ⇒ We modified figure S1, including images of the actual device and microscope images of the flow lines filled with a fluorescent dye. We also included a video, illustrating simple chemostat operations.
 - ⇒ A detailed description of the chip, typical experimental operation of the chip and essential operations and characterization of the chip are given in the cited protocol (Laohakunakorn, 2021, in press), and currently available as a preprint (Laohakunakorn, 2020).
- For most time course graphs on figures 2, 3 and 4: the traces do not have error bars.
 - ⇒ That is correct, our time course graphs do not have error bars. Since we only have a small number of technical replicates for each condition, we opted to show both traces rather than an average + error bar, as the experimental variability and reproducibility is then clearly shown to the reader. (see similar response to reviewer #1)
- Could you, please, explain what does the “kickstart” phase of the reaction mean?
 - ⇒ We believe that we the different phases including the kickstart phase sufficiently in the manuscript.
 - ⇒ Page 7, Line 80, 83 and 86: “We developed a ‘kick-start’ method to enable the system to self-regenerate proteins from DNA templates (Fig. 1C)... The kick-start phase is required to allow a productive switch from a complete to a Δ PURE system to occur. In the kick-start phase, which lasts for the first 4h, linear DNA templates coding for eGFP and the protein to be regenerated are added to a complete PURE system. This leads to the expression of eGFP and the protein to be regenerated.”
- On many time course traces on figures 2, 3 and 4, there appears to be GFP signal coming up as soon as ~half an hour (I’m approximating in the time between 0 and 5h labeled on axis). This is extremely fast for a signal from GFP in PURE system. Maturation time of eGFP variant (Kremers 2007) is 128 minutes, and best new mutants I’ve seen mature as fast as little over an hour. Do Authors have a particularly fast maturing variant?

- ⇒ For our experiments, we used the eGFP variant with the same mutations as in Kremers 2007. We do detect fluorescence signal within 10-30 min of the experiment, which is in agreement with our batch experiments (Supplementary Fig. S2). The times observed are consistent with other literature reports using eGFP (Niederholtmeyer 2013, Lavickova 2019) and other GFP variants (Kazuta 2014, Villarreal 2018).
- ⇒ The observation of signal and timing also relates to the sensitivity of experimental setup including the camera/detector used.
- ⇒ Furthermore, maturation times reported in literature for eGFP vary widely. The reviewer referenced a paper by Kremers, where the maturation time was measured in-vivo. However, more recent findings for maturation time of eGFP in-vivo (Balleza, 2018) and in-vitro in the PURE system (Iizuka 2011, Niederholtmeyer 2013), measured much faster maturation times (10-15 min).
- The reader is referred to [protocols.io](https://www.protocols.io) for the details of the construction of microfluidic rig and imaging system. As far as I know, [protocols.io](https://www.protocols.io) is not peer reviewed. In fact, even relying on unsecured community voting system to validate protocols, this protocol only has 1 endorsement (and that's from one of the authors). I would suggest that such crucial part of the paper as preparation of novel hardware and data collection should be included in the methods for this manuscript, to be reviewed by a subject matter expert before publication.
 - ⇒ We agree with the reviewer that this is a crucial part of the manuscript. Therefore, we believe that the scientific community will benefit from the additional information provided on protocols.io. Moreover, the protocol published on protocols.io is a “pre-print” for a peer-reviewed protocol, which was accepted during the review process for this manuscript and is currently in press for the volume 2229 of Methods in Molecular Biology.
 - ⇒ We revised the manuscript and updated the references accordingly.
 - ⇒ Revised reference: “Nadanai Laohakunakorn, Barbora Lavickova, Zoe Swank, Julie Laurent, and Sebastian J. Maerkl. Steady-state cell-free gene expression with microfluidic chemostats. In Filippo Menolascina, editor, Synthetic Gene Circuits, volume 2229 of Methods in Molecular Biology. Springer US, 1 edition, 2021.”

The synthesis of T7 RNA polymerase from a gene encoded under T7 promoter was demonstrated before. This is a very interesting part of the paper, and the resource allocation discussion is interesting and certainly could be expanded. This is, however, not an example of self-regeneration. Minor point: half-life of T7 RNAP activity would be useful to demonstrate at which time point the replacement of endogenously synthesized T7RNAP is complete. In my opinion, there is technical novelty in this paper, and the problem of resource allocation in cell-free transcription and translation systems is definitely timely and worth studying. However, the paper uses “regeneration” systems that has been demonstrated before (T7 RNAP synthesis under T7 promoter), and the conclusions made in this paper are too general.

- ⇒ Unfortunately, the reviewer did not provide any specific literature references in support of any of the comments, which makes it difficult to validate or directly respond to the claims made by the reviewer. We do not claim that T7 RNAP was not synthesized under control of a T7 promoter previously. We did our best to list all the relevant studies in this field and report on all prior work relevant to this topic. However, we are not aware of any paper that demonstrates long-term synthesis and regeneration of T7 RNAP in a minimal in vitro transcription – translation system. Nor are we aware of any papers that show the same for one or more aaRSs. RNAP is a key component of TX-TL systems, therefore it will need to be regenerated to achieve fully self-regenerating systems (or a synthetic cell).

- ⇒ The negative control shows the half-life of T7RNAP in the system (Fig. 3C) and is precisely the reason we included this control. We think that our negative control is the most accurate representation for the decrease in T7 RNAP activity as not only the T7 RNAP has to be replaced, but also any DNA template and T7 RNAP mRNA produced has to be washed out of the system.
- ⇒ We address the comment about the use of the term self-regeneration and conclusions below.
- The authors study in great detail regeneration of few key elements of PURE system, but they do not provide rationale of how those studies can help build completely self-regenerating cell-free protein expression system, and consequently a synthetic minimal cell.
 - ⇒ Our system demonstrates that the PURE system is capable of synthesizing a number of components functionally and at sufficient concentration to maintain sustained steady-state activity. So far, we successfully regenerated 7 aaRSs and T7RNAP, the platform is general enough to be used to eventually test and hopefully regenerate all components of the PURE system individually and in combination. It is thus a general and powerful approach to experimentally characterize the steps required to achieve complete self-regeneration. Additionally, we identified factors such as resource competition/allocation, which are general principles which can help in the construction of regenerating systems. We would go as far as to argue that this is the most likely approach to lead to successfully develop a biochemical universal constructor and ultimately a synthetic cell. It is to our knowledge the first demonstration of this strategy towards the development of a synthetic cell, with all other efforts seeming to focus on developing vesicle-based systems or the in vitro synthesis and reconstitution of proteins in simple manual batch processes.
- Overall, the paper is technically sound and represents great experimental system, worked out with impressive details.
 - ⇒ We thank the reviewer for the recognition that our work “is technically sound and represents a great experimental system”.
- However, the conclusions made in the paper are not supported by the experimental data. Calling the system self-regenerating is an overstatement, and there is not enough of evidence to sufficiently back up a claim that this is a general model for self-regenerating synthetic cell.
 - ⇒ We respectfully disagree with these opinions. We also find it difficult to further discuss these points as the reviewer didn’t support these opinions with specific examples, fact-based arguments, or references to literature.
 - ⇒ We in fact spent a significant amount of time discussing exactly these points, and placed a lot of effort on finding and using the appropriate terms to describe the system we developed. We do not claim that the system is completely self-replicating. That would indeed be an overstatement. We furthermore specifically use the term self-regeneration because it doesn’t imply complete system regeneration as would have been the case if one would have used the term self-replication for example. The term self-regeneration applies to partial as well as complete self-regeneration, whereas the term self-replication requires complete self-regeneration of a system. For example, using the title: “A self-replicating synthetic cell model” and certainly “A self-

replicating synthetic cell” would be overstatements. Using the title “A self-regenerating synthetic cell model” is in our opinion an accurate summary description of our work.

- ⇒ In the paper, we do not state that the system itself is completely self-regenerating (we do not use terms like fully or completely self-regenerating), but we claim and show that it self-regenerates some of its components (see comments above for similar remarks in regards to the term self-regeneration). As stated above, we identify factors which are important for any attempts at achieving complete self-regeneration or self-replication and we demonstrate a platform which can be used to assess the success of self-regeneration (and can eventually be used to show full self-regeneration). Therefore, the conclusions made in this paper are important for progress towards full self-regeneration.

Reviewers' Comments:

Reviewer #1:

Remarks to the Author:

The authors are well responding to the suggestions by the reviewer and these have been reflected in the revised manuscript.

Therefore, the reviewer believe there is no problem to publish in the Nature communications.

Reviewer #2:

Remarks to the Author:

I'm still not entirely happy about the distinction Authors make about 'synthetic cell' vs 'synthetic cell model'. Every synthetic cell is a model cell right now, since we have none that are really alive.

So, the word "model" is redundant, and the paper title is effectively claiming to have self regenerating synthetic cell - which the paper does not show.

Its also not truly self-regenerating (as in: does not replenish all its key components).

The Authors addressed all other comments and questions.